# Rough-Set-Theory-Based Classification with Optimized *k*-Means Discretization

**Teguh Handjojo Dwiputranto \*, Noor Akhmad Setiawan 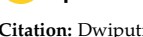 and Teguh Bharata Adji**

Department of Electrical and Information Engineering, Universitas Gadjah Mada, Yogyakarta 55281, Indonesia; noorwewe@ugm.ac.id (N.A.S.); adji@ugm.ac.id (T.B.A.)
\* Correspondence: teguh.handjojo.d@mail.ugm.ac.id; Tel.: +62-822-8480-1480

**Abstract:** The discretization of continuous attributes in a dataset is an essential step before the Rough-Set-Theory (RST)-based classification process is applied. There are many methods for discretization, but not many of them have linked the RST instruments from the beginning of the discretization process. The objective of this research is to propose a method to improve the accuracy and reliability of the RST-based classifier model by involving RST instruments at the beginning of the discretization process. In the proposed method, a *k*-means-based discretization method optimized with a genetic algorithm (GA) was introduced. Four datasets taken from UCI were selected to test the performance of the proposed method. The evaluation of the proposed discretization technique for RST-based classification is performed by comparing it to other discretization methods, i.e., equal-frequency and entropy-based. The performance comparison among these methods is measured by the number of bins and rules generated and by its accuracy, precision, and recall. A Friedman test continued with post hoc analysis is also applied to measure the significance of the difference in performance. The experimental results indicate that, in general, the performance of the proposed discretization method is significantly better than the other compared methods.

**Keywords:** rough set theory; genetic algorithm; discretization; classification; data pre-processing



## 1. Introduction

Classification is one of the processes commonly completed by researchers in machine learning (ML). In general, the purpose of classification is to assign an object to one of the categories that has been predefined. Currently, there are various algorithms for classification, such as Decision Tree, Artificial Neural Network, Random Forest, Fuzzy Logic, and many more, including Rough Set Theory (RST). To obtain the best result, selecting the proper algorithm is crucial by considering not only the accuracy but also the cost of training, cost of testing, and cost of the implementation. Another important factor is whether the classification model needs to be built as a white or black box model. If a white box model is expected, a method such as Decision Tree, Fuzzy Logic, or RST can be applied because this method can produce transparent decision rules.

In a dataset that will be processed for classification, attributes that have continuous values are often found. Hence, the data of the attributes cannot be directly processed by a classifier that requires discrete data, such as RST. To be able to process the dataset, a discretization process should be carried out first.

Currently, there are many state-of-the-art methods for discretization, as reported in Refs. [1,2]. Based on this report, there are two main groups of discretization methods, i.e., supervised and unsupervised. This work also conducted a survey, finding that the popular methods for unsupervised discretization use an equal-width and equal-frequency base. The disadvantage of this unsupervised method is that we cannot be sure whether the discrete results are optimal since there is no feedback to measure the optimality of discrete results at the time of the process. To generate optimal discretized values, a supervised

method should be applied. One of the popular methods for supervised discretization is entropy-based [1]. However, the next question is whether the entropy-based method will be suitable or not for RST-based classifiers.

This paper aims to improve the classification performance using the RST method on various datasets with continuous values obtained from UCI. The contribution of this study is to propose data pre-processing methods related to discretization before carrying out the classification process. The proposed method starts with applying *k*-means to discretize continuous value attributes, then optimizes them by using a genetic algorithm (GA) that involves one of the RST instruments, called the dependency coefficient, to maintain the quality of the dataset as the original after the implementation of the discrete process.

By involving one of the RST elements in the discretization process, it is expected that the discretization results will be suitable for the RST-based classifier. Thus, the novelty of the proposed method compared to other discretization processes is that the method is based on approximation quality with the expectation that it will give better results to be used by the RST-based classifier because the approximation is controlled by one of the RST elements from the beginning.

This paper is organized as follows: Section 2 explains the theoretical basis of the RST, which begins with the concept of approximation in the framework of rough sets, and then continues with an explanation of the basic notions and characteristics of the RST. Section 3 presents the need for discretization and its various techniques, especially those related to the proposed method. Section 4 describes the basic concepts of the proposed method and the algorithm in pseudo-code form. Section 5 presents the experimental framework, the datasets used, and other popular discretization methods. Section 6 describes the analysis of the experimental results, and this paper is concluded in Section 7.

## 2. Basic Notions

Before the detailed description of the method proposed in this article is discussed, a basic picture of RST that was first proposed by Zdzislaw Pawlak in 1982 will be given. This RST method is intended to classify and analyze imprecise, uncertain, or incomplete information and knowledge [3,4]. The underlying concept of the RST is the size approximation of the lower and upper sets. The approximation of the size of the lower subset is determined by the group of objects that are becoming members of the desired subset. Meanwhile, the size of the upper subset approximation is determined by the possible group of objects to become a member of the desired subset. Any subset defined or bordered by an upper–lower approximation is called a Rough Set [3]. Since it was proposed, RST has been used as a valuable tool for solving various problems, such as for imprecise or uncertain knowledge representation, knowledge analysis, quality measurement of the information available on the data pattern, data dependency and uncertainty analysis, and information reduction [5].

This RST approach also contributes to the artificial intelligence (AI) foundation, especially in machine learning, knowledge discovery, decision analysis, expert systems, inductive reasoning, and pattern recognition [3].

The rough sets approach has many advantages. Some of the most prominent advantages of applying RST are 6:

1.　Efficient in finding hidden patterns in the dataset;
2.　Able to identify difficult data relationships;
3.　Able to reduce the amount of data to a minimum (data reduction);
4.　Able to evaluate the level of significance of the data;
5.　Able to produce a set of rules for transparent classification.

The following sub-sections will explain the basic and important philosophies associated with RST to be discussed based on Refs. [3,6–9].

*2.1. Equivalent Relations*

Let $U$ be a non-empty set, whereas $p$, $q$, and $r$ are elements of $U$. If $R$ is a symbol of a relation so that $pRq$ is a relation function between $p$ and $q$, then $R$ is said to be an equivalent relation when it meets three properties as follows:

1. Reflexive: $pRp$ for all $p$ in $U$;
2. Symmetric: if $pRq$, then $qRp$;
3. Transitive: if $pRq$ and $qRr$, then $pRr$.
4. If $x$ in $U$, then $Rx = \{y \in U : yRx\}$ is the equivalence class of $x$ with respect to $R$.

*2.2. Information System and Relationship Indiscernibility*

Let $T = (U, A, Q, \rho)$ be an Information System ($IS$), where $U$ is a set of non-empty objects called universe, $A$ is a set of attributes, $Q$ is the union among the attribute domains in $A$, and $\rho : U \times Q \to A$ is the description of the total function. For classification, the set of attributes, $A$, is divided into condition attributes denoted by $CON$ and a decision attribute denoted by $DEC$. When the attributes of the information table have been divided into condition and decision attributes, then the table is called a decision table. The element of $U$ can be called object, case, instance, or observation [10]. The attributes can be called features, variables, or characteristic conditions. If an attribute $a$ is given, then: $a : U \to V_a$ for $a \in A$. $V_a$ is called the set of values of $a$.

If $a \in A$, $P \subseteq A$, then an indiscernibility relation $IND(P)$ can be defined as: $IND(P) = \{(x,y) \in U \times U : \text{for all } a \in P, a(x) = a(y)\}$, or in the statement that the two objects are said to be indiscernible when the two objects are indistinguishable since they do not have sufficient differences in the set of attributes called $P$. The equivalence class of indiscernibility relation $IND(P)$ is denoted by $[X]_P$.

*2.3. Lower Approximation Subset*

Let $B \subseteq C$, where $C$ is a set of condition attributes, and $X \subseteq U$; then, the $B$-lower approximation subset of $X$ is the set of all elements of $U$ that can be classified exactly as an element of $X$, and it is shown in Equation (1):

$$B_*(X) = \{x \in U : [X]_B \subseteq X\} \tag{1}$$

*2.4. Upper Approximation Subset*

A $B$-upper approximation subset of $X$ is the set of all elements of $U$ that may be classified as elements of $X$, and this is shown in Equation (2):

$$B^*(X) = \{x \in U : [X]_B \cap X \neq \varnothing\} \tag{2}$$

*2.5. Boundary Region Subset*

This subset contains a group of elements as defined in Equation (3). This set contains objects that, whether they belong to the $X$ classification, cannot be determined exactly.

$$BN_B(X) = B^*(X) - B_*(X) \tag{3}$$

*2.6. Rough Set*

A set obtained by the lower and upper approximations is called a rough set. When a rough set is found, then it must be $B^*(X) \neq B_*(X)$. Figure 1 illustrates each set that meets Equations (1)–(3).

*2.7. Crisp Set*

If $B^*(X) = B_*(X)$, then the set is called a crisp set.

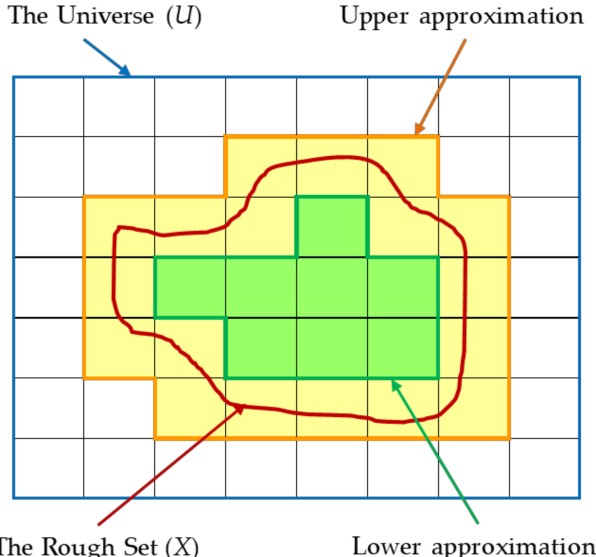

**Figure 1.** The illustration of rough set. The universe ($U$) is the union of all blocks. If the set ($X$) is represented by the red shape, then the lower approximation is the union of all green blocks, the upper approximation is the union of all green and yellow blocks, and the boundary region is the union of yellow blocks, while the union of all white blocks is called outside approximation.

### 2.8. Positive Region Subset

This is a set that has an object of the universal set $U$ that can be classified or partitioned into certain classes of $U/D$ using the set of attributes $C$, as shown in Equation (4).

$$POS_C(D) = \bigcup C_*(X), \tag{4}$$

where $U/D$ is the partitioning of $U$ based on the attribute values of $D$ and $C_*(X)$ is the notation of lower approximation of the set $X$ with respect to $C$. The positive region of the subset $X$ belonging to the partition $U/D$ is also called the lower approximation of the set $X$. The positive region of a decision attribute with respect to a subset $C$ approximately represents the quality of $C$. The union of the positive and the boundary regions yields the upper approximation [7].

### 2.9. Dependency Coefficient

Let $T = (U, A, C, D)$ be a decision table. The dependency coefficient between attribute condition $C$ and attribute decision $D$ can be formulated as in Equation (5) as follows:

$$\gamma(C, D) = |POS_C(D)|/|U| \tag{5}$$

The value of the dependency coefficient is in the range from 0 to 1. This coefficient represents a portion of the objects that can be correctly classified against the total. If $\gamma = 1$, then $D$ is completely related to $C$, if $0 < \gamma < 1$, then $D$ is said to have partial relation on $C$, and if $\gamma = 0$, then $D$ has no dependency to $C$. A decision table depends on the feature set condition when all values on the decision feature $D$ can be uniquely determined by the condition attribute values.

### 2.10. Reduction of Attributes

As explained in Section 2.2., it is possible that two or more objects are indiscernible because they do not have enough different attribute values. In this case, it is necessary to make savings so that only one element of the equivalence class is required to represent the whole class. To be able to make savings, some additional notions are needed.

Let $T = (U, A)$ be an information system, $P \subseteq A$, and let $a \in P$. It can be said that a is dispensable in $P$ if $IND_T(P) = IND_T(P - a)$; otherwise, $a$ is indispensable in $P$. A set $P$ is called independent if all of its attributes are indispensable.

Any subset $P'$ of $P$ is called a reduct of $P$ if $P'$ is independent and $IND_T(P') = IND_T(P)$.

Therefore, *reduct* is the minimal set of attributes without changing the classification results when using all attributes. In other words, the attributes not in *reduct* are considered redundant and have no effect on classification.

### 2.11. Discernibility Matrix and Function

*Reducts* have several properties, one of which is the validity of the relation, as shown in Equation (6). Let $P$ be a subset of $A$. The *core* of $P$ is the set off all indispensable attributes of $P$ [10].

$$Core(P) = \bigcap Red(P), \tag{6}$$

where $Red(P)$ is the set of all *reducts* of $P$.

In order to easily calculate *reduct* and *core*, discernibility matrix can be used [10], which is defined as follows.

Let $T = (U, A)$ be an information system with $n$ objects. The discernibility matrix of $T$ is a symmetric $n \times n$ matrix with entries in $c_{ij}$, as given in Equation (7).

$$c_{ij} = \left\{ a \in A \middle| a(x_i) \neq a(x_j) \right\} \text{ for } i, j = 1, \ldots, n \tag{7}$$

A discernibility function $f_T$ for an information system $T$ is a Boolean function of $m$ Boolean variables $a_1^*, \ldots, a_m^*$ (corresponding to the attribute $a_1, \ldots, a_m$), defined as follows:

$$f_T(a_1^*, \ldots, a_m^*) = \forall \left\{ \exists c_{ij}^* \middle| 1 \leq j \leq i \leq n, c_{ij} \neq \varnothing \right\}, \tag{8}$$

where $c_{ij} = \left\{ a^* \middle| a \in c_{ij} \right\}$.

## 3. Discretization

Discretization is one of the data preprocessing activity types performed in the preparation stage as well as data normalization, data cleaning, data integration, and so on. Often, data preprocessing needs to be performed to improve the efficiency in subsequent processes [11]. It is also needed to meet the requirements of the method or algorithm to be executed. The rough-set-theory-based method is one of the methods that requires data in the discrete form. Therefore, if the dataset to be processed is in continuous mode, then the discretization process is required.

There are several well-known discretization techniques that can be categorized based on how the discretization process is carried out. When it is carried out by referring to the labels that have been provided in the dataset, then it is called supervised discretization, while, if the label is not available, then it is categorized as unsupervised discretization [11].

Discretization by binning is one of the discretization techniques based on a specified number of bins. If the dataset has a label, then the number of bins for discretization can be determined for as many as the number of classes on the label, while, for a dataset with no label, an unsupervised technique, such as clustering, should be applied.

### 3.1. k-Means

Cluster analysis or clustering is one of the most popular methods for discretization. This technique can be used to discretize a numeric attribute, $A$, by dividing the values of $A$ into several clusters [11]. This experiment applies the *k*-means method to discretize the numeric attributes of the dataset.

*k*-means is a centroid-based method. Assume $A$ is one of the numeric attributes of a dataset $D$. Partitioning can be performed on the $A$ attribute into $k$ clusters, $C_1, C_2, \ldots, C_k$, where $C_i \subset A$ and $C_i \cap C_j = \varnothing$ for $(1 \leq i, j \leq k)$. In *k*-means, the centroid, $c_i$, of a cluster $C_i$ is the center point that is defined as the mean of the points assigned to the cluster. The

difference between a point, $p_n$, and its centroid, $c_i$, is measured using a distance function, $dist(p_n, c_i)$. The most popular formula to measure the distance is by using the Euclidean distance formula, as shown by Equation (9).

$$dist(x, y) = \sqrt{\sum_{i=1}^{n} (x_i - y_i)^2} \tag{9}$$

Because *k*-means is one of the unsupervised techniques, then the value of *k* is not known and it is usually defined through trial and error iteratively to find the optimum value. To automate this trial-and-error process, an optimization technique should be applied. There are many optimization techniques available, but this experiment employs genetic algorithm (GA) technique to find the optimum value for *k*.

In this experiment, *k* is optimum if the value is as minimal as possible without losing the quality of the information of the dataset. This experiment uses $\gamma(C, D)$ function, as shown in Equation (5).

### 3.2. Genetic Algorithm

Genetic algorithm (GA) is an algorithm inspired by biological phenomena, namely the process of genetic evolution from the creation of a population that consists of some individuals who later experience genetic evolution. There are three genetic processes that occur, i.e., selection, crossover, and mutation, to obtain new individuals who are expected to be stronger or fitter during the next cycle selection process [12]. Figure 2 shows GA's operational processes. Figure 3 illustrates the crossover process.

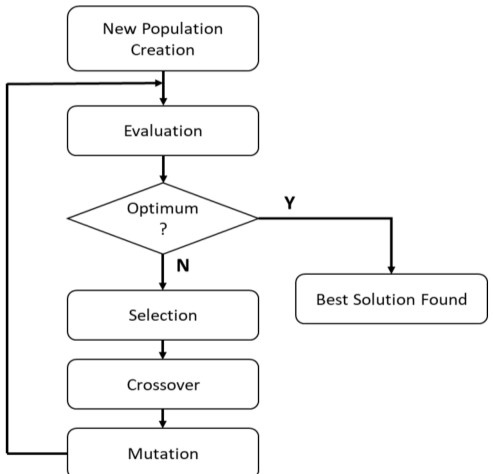

**Figure 2.** Genetic algorithm process flow.

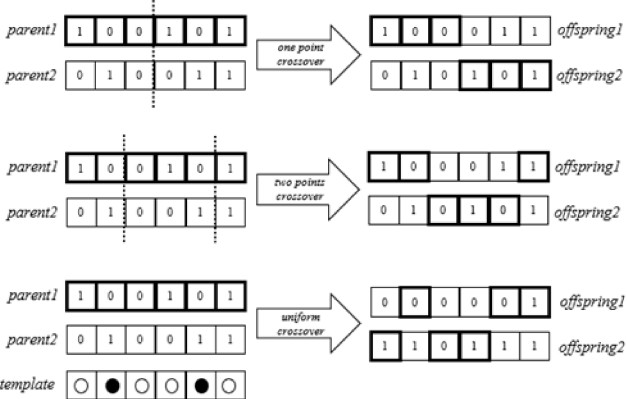

**Figure 3.** Illustration of some crossover-type processes.

## 4. Proposed Method

The concept of the proposed method for the discretization in this experiment is the integration of RST, *k*-means, and GA. An RST is used to measure the dependency coefficient, which can be used to define the approximation quality. Therefore, the transformed dataset after the discretization process will not decrease the quality of the information from the original dataset. To measure the approximation quality, the formula of RST dependency coefficient, $\gamma(C, D)$, as shown in Equation (5), is applied.

Further, *k*-means is applied to cluster continuum data attributes. The result is the number of bins or clusters of the attributes. The bins are then transformed into discrete values. The GA function is used to minimize the number of bins or clusters of every attribute, which, at the same time, must meet the constraint in which the value of $\gamma(C, D)$ is equal to 1 or any value that is targeted. Minimizing the number of bins is expected to generate the most optimum number of RST rules, which make the classification process become more efficient. The following algorithm of the proposed method is developed to find the most optimal discretization scenario of an Information System.

As shown on the *pseudo-code*, the algorithm of the proposed method begins with reading the training dataset to construct a table called $T = (U, A, V, f)$, where $U$ is a set of objects, $A$ is a set of the attributes, $V$ is a set of values of the attributes, and $f$ is a function of the relationship between the object and the attributes. This table is then transformed into a decision table, called $DT = (U, C, D, v, f)$, where $C$ is the condition attribute set and $D$ is the decision attribute set that satisfies $C \cup D = A$.

After the dataset is loaded, the process continues with the setting of the GA process, starting from the number of chromosomes, which is associated with the number of attributes, and followed by the number of genes for each chromosome, which is associated with the number of centroids or bins of the respective attribute. After the setting of the GA parameters is completed, it continues by executing the GA processes based on Figures 2 and 3. The objective function of the GA is to minimize the number of bins for each attribute with a certain value of $\gamma(C, D)$ as the constraint.

The end of the GA iteration contains the process to convert the chromosome values into the attribute bin values. When the maximum iteration is achieved, then the bin values of each attribute are considered optimum and then are used to discretize the condition of attribute values.

## 5. Experimental Setup

In this section, the test results of the proposed algorithm are compared with two popular discretization algorithms, namely *equal-frequency*, which is processed using unsupervised learning, and *entropy-based*, which uses supervised learning.

Four datasets downloaded from the UCI data repository with details of the properties owned by each dataset shown in Table 1 are selected. Those datasets are:

1.   iris;
2.   ecoli;
3.   wine;
4.   banknote.

**Table 1.** Descriptions of the tested datasets in this research.

| Properties | Datasets | | | |
| --- | --- | --- | --- | --- |
| | *iris* | *ecoli* | *wine* | *banknote* |
| # of examples | 150 | 271 | 178 | 1370 |
| # of classes | 3 | 8 | 3 | 2 |
| # of condition attributes | 4 | 7 | 13 | 4 |

The proposed algorithm was tested on four datasets and compared with two discretization methods, namely *equal-frequency* and *entropy-based*. Figure 4 shows the flow of the research.

In the initial step, a *k*-fold mechanism with $k = 5$ is applied to each dataset so that a ratio of 80:20 is obtained, where 80% of the data are used for the training and 20% for testing. The *k*-fold approach is applied to ensure that every record in the dataset becomes either a training or test dataset. With the application of *k*-fold, it is expected that the results of testing the algorithm can be more reliable. Each fold of each dataset is then discretized using three tested methods, namely: *equal-frequency* (EQFREQ), *entropy-based* (ENTROPY), and the proposed method, which is based on genetic algorithm and rough set theory (GARST).

Discretization with the EQFREQ and ENTROPY methods was concluded on the Rosetta software ver. 1.4.41. Meanwhile, the proposed method was developed by using Python 3.8 based on Algorithm 1.

---

**Algorithm 1.** *Pseudo-code* of proposed method.

---

**Input:**       A dataset in the form of Table $T = (U, A, V, f)$
**Output:**      Optimum numbers of bins for each condition attribute in the form of discretized table $DiscT = (U, C, D, V_c \text{ disc}, f)$
Create decision table $DT = (U, C, D, V, f) = convert\_to\_DT(T)$, where $C \cup D = A$;
Introduce integer variable $maxK = 10$ or any integer value;
Introduce scalar and vector variables *genBit, numChrom, popSize, max Generation, constraintGA,*
   *Chromosome, Individu, Fitness, Parents, Offsprings, New Pop* for the GA processes;
$genBit \leftarrow integer\_to\_bineary(maxK);$ $numChrom \leftarrow cardinality(C);$
$popSize \leftarrow 30$ or any integer value;
$maxGeneration \leftarrow 50$ or any integer value;
**for** $indv \leftarrow 1$ **to** $popSize$ **do**
   **for** $chr \leftarrow 1$ **to** $numChrom$ **do**
      $Chromosome[chr] \leftarrow binary\_random(genBit);$
   **end**
$Individu[indv] \leftarrow [Chromosome[numChrom]];$
**end**
$constraintGA \leftarrow 0.8$ or any real value between 0.0 and 1.0;
Introduce vector variables *Bins, Discr_V,* $\gamma CD$ for the RST processes;
**for** generation $\leftarrow 1$ **to** $maxGeneration$ **do**
   **for** $indv \leftarrow 1$ **to** $popSize$ **do**
      **for** $chr \leftarrow 1$ **to** $numChrom$ **do**
$Bins[chr] = KMeans(C[chr], binary\_to\_integer(Individu[chr]));$
**End**
 **for** c $\leftarrow 1$ **to** *cardinality* **do**
   $Discr\_V[c] \leftarrow discretize(V[c], Bins[c]);$
   $\gamma CD[indvc] \leftarrow calc\_\gamma CD(Discr\_V[c], V[dc])$ by referring to Eq. 2.5;
   **if** $\gamma CD[indvc] \geq constraintGA$ **then**
      $Fitness[indv] \leftarrow sum\_cardinality(Bins[1], \ldots, Bins[numChrom])$
   **else** $Fitness[indv] \leftarrow very\_big\_vaule;$
**End**
**End**
$Parents \leftarrow select\_the\_most\_fit(Individu[1], \ldots, Individu[popSize])$ to create parents; the
   *Individu* have smaller *Fitness* value will have chance to be selected as a parent;
$Offsprings \leftarrow crossover(Parents)$ to create *Offsprings*;
$NewPop \leftarrow mutate(Offsprings);$
Run $transform(NewPop)$ to create new list of *Individu* in the form of
$[Individu[1], \ldots, Individu[popSize]];$
**end**
**return** $DiscT = (U, C, D, V, disc, f)$

---

After the *5*-fold datasets have been discretized, each fold is reduced and then rules generation is performed using the Rosetta software. The *reduct* process is carried out using the RST method based on a discernibility matrix, and rule generation using the application of Boolean algebra to the built discernibility matrix, as described in Section 2. This process is repeated five times for each dataset due to the application of *5*-fold.

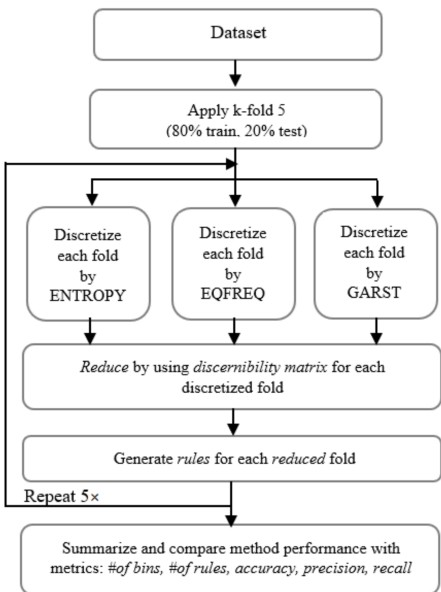

**Figure 4.** Flow of the research.

The final step of this experiment is to compare the performance of the three methods. The measuring instruments used in the experiment and their explanations are listed in Table 2.

**Table 2.** Metrics to measure the performance.

| Measurement Unit | Objective | Remarks |
|---|---|---|
| # of bins | An integer value that indicates the number of bins resulting from discretization. | The smaller this value, the better the performance of the discretization method because the dataset resulting from the discretization becomes simpler. |
| # of rules | An integer value that indicates the number of rules generated by RST after the *reduct* process. | The smaller this number indicates the better performance of the discretization method because the smaller number of rules makes it easier to understand and more transparent. |
| Accuracy | Provides a measure of how many samples were correctly predicted by a classifier compared to the total number of samples. | This metric is applied to measure the overall performance. |
| Precision | Provides a measurement of how many samples are correctly predicted for a particular class. This is the TP ratio of a given class to the number of samples predicted as this class, in other words, the total number of TP and FP. | This metric is applied to measure the class-by-class performance of a method. |
| Recall | Provides a measurement of how many samples are correctly predicted in a given class. | This metric also measures the class-by-class performance of a model. |

To ensure that there is a difference in performance between the three tested methods, the statistical Friedman test method was applied to this experiment. The Friedman test is a statistical measuring tool used to determine whether there is a statistically significant difference in the average value of three or more groups [13]. If the *p*-value of the Friedman

test is less than 0.05, then there is a significant difference. The post hoc test was used as a continuation of the Friedman test to determine which group had a significant difference compared to the other groups.

## 6. Results and Discussion

After the entire process is completed, the last step is to review the performance of each discretization method. Table 3 shows the performance comparison of the discretization methods of the *equal-frequency* (EQFREQ), *entropy-based* (ENTROPY), and genetic algorithm and rough set theory (GARST) proposed in this paper.

**Table 3.** Number of bins and rules generated by each method.

| | | iris | | ecoli | | wine | | banknote | |
|---|---|---|---|---|---|---|---|---|---|
| | | # of bins | # of rules | # of bins | # of rules | # of bins | # of rules | # of bins | # of rules |
| ENTROPY | Fold-1 | 20 | 49 | 42 | 104 | 143 | 3952 | 501 | 2941 |
| | Fold-2 | 20 | 60 | 36 | 107 | 152 | 4874 | 296 | 341 |
| | Fold-3 | 21 | 83 | 32 | 137 | 155 | 6892 | 454 | 1364 |
| | Fold-4 | 21 | 87 | 45 | 116 | 164 | 8366 | 564 | 2760 |
| | Fold-5 | 21 | 102 | 43 | 118 | 167 | 9394 | 498 | 1423 |
| | *Average* | *20.6* | *76.2* | *39.6* | ***116.4*** | *156.2* | *6695.6* | *462.6* | *1765.8* |
| | *Max* | *21* | *102* | *45* | *137* | *167* | *9394* | *564* | *2941* |
| | *Min* | *20* | *49* | *32* | *104* | *143* | *3952* | *296* | *341* |
| | *StdDev* | *0.4899* | *19.1353* | *4.8415* | *11.5689* | *8.6116* | *2047.2293* | *90.3761* | *967.3199* |
| EQFREQ | Fold-1 | 20 | 186 | 27 | 401 | 65 | 50473 | 20 | 158 |
| | Fold-2 | 20 | 192 | 27 | 218 | 65 | 48770 | 20 | 149 |
| | Fold-3 | 20 | 220 | 27 | 214 | 65 | 49921 | 20 | 154 |
| | Fold-4 | 20 | 135 | 27 | 215 | 65 | 49929 | 20 | 157 |
| | Fold-5 | 20 | 186 | 27 | 211 | 65 | 51401 | 20 | 151 |
| | *Average* | *20* | *183.8* | ***27*** | *251.8* | *65* | *50098.8* | *20* | *153.8* |
| | *Max* | *20* | *220* | *27* | *401* | *65* | *51401* | *20* | *158* |
| | *Min* | *20* | *135* | *27* | *211* | *65* | *48770* | *20* | *149* |
| | *StdDev* | *0.0000* | *27.4547* | *0.0000* | *74.6335* | *0.0000* | *855.7926* | *0.0000* | *3.4293* |
| GARST | Fold-1 | 17 | 39 | 31 | 164 | 46 | 2527 | 17 | 55 |
| | Fold-2 | 11 | 54 | 29 | 277 | 23 | 2995 | 13 | 36 |
| | Fold-3 | 13 | 21 | 28 | 154 | 43 | 5212 | 13 | 73 |
| | Fold-4 | 16 | 25 | 29 | 136 | 45 | 10319 | 15 | 88 |
| | Fold-5 | 11 | 53 | 30 | 134 | 46 | 5734 | 13 | 65 |
| | *Average* | ***13.6*** | ***38.4*** | *29.4* | *173* | ***40.6*** | *5357.4* | ***14.2*** | ***63.4*** |
| | *Max* | *17* | *54* | *31* | *277* | *46* | *10319* | *17* | *88* |
| | *Min* | *11* | *21* | *28* | *134* | *23* | *2527* | *13* | *36* |
| | *StdDev* | *2.4980* | *13.7055* | *1.0198* | *53.1940* | *8.8679* | *2770.2903* | *1.6000* | *17.4425* |

Compared to the performance of the EQFREQ and ENTROPY discretization methods, it is confirmed that the proposed method (GARST) has a better performance, showing the smallest number of the generated bins and rules across three datasets, namely *iris*, *wine*, and *banknote*. The ENTROPY method indicates a better performance for the *ecoli* dataset, demonstrated by the smallest number of bins; however, the GARST method is still superior because it succeeded in generating the smallest number of rules in all the datasets, including *ecoli*.

Table 4 shows the test results that are presented in statistical measures, namely average and standard deviation. From this table, it can be seen that the GARST method has the highest average *accuracy*, *precision*, and *recall*, and has competitive values for standard deviation.

**Table 4.** The *accuracy*, *precision*, and *recall* of each method.

|  |  | *iris* | | | *ecoli* | | | *wine* | | | *banknote* | | |
|---|---|---|---|---|---|---|---|---|---|---|---|---|---|
|  |  | Acc | Avg | Avg | Acc | Avg | Avg | Acc | Avg | Avg | Acc | Avg | Avg |
|  |  | (%) | Prec | Recall | (%) | Prec | Recall | (%) | Prec | Recall | (%) | Prec | Recall |
| ENTROPY | Fold-1 | 96.67 | 0.97 | 0.95 | 29.85 | 0.18 | 0.21 | 38.89 | 0.42 | 0.41 | 74.40 | 0.74 | 0.74 |
|  | Fold-2 | 93.33 | 0.93 | 0.93 | 34.33 | 0.27 | 0.33 | 50.00 | 0.50 | 0.51 | 99.54 | 0.81 | 0.81 |
|  | Fold-3 | 96.67 | 0.97 | 0.97 | 26.87 | 0.42 | 0.40 | 41.67 | 0.38 | 0.40 | 99.89 | 0.84 | 0.84 |
|  | Fold-4 | 93.33 | 0.94 | 0.94 | 20.90 | 0.24 | 0.19 | 50.00 | 0.44 | 0.47 | 99.88 | 0.65 | 0.65 |
|  | Fold-5 | 93.33 | 0.95 | 0.95 | 25.37 | 0.23 | 0.20 | 49.65 | 0.51 | 0.49 | 99.96 | 0.83 | 0.82 |
|  | *Global Avg* | *94.67* | ***0.95*** | *0.95* | *27.46* | *0.27* | *0.27* | *46.04* | *0.45* | *0.46* | *94.73* | *0.77* | *0.77* |
|  | *Max* | *96.67* | *0.97* | *0.97* | *34.33* | *0.42* | *0.40* | *50.00* | *0.51* | *0.51* | *99.96* | *0.84* | *0.84* |
|  | *Min* | *93.33* | *0.93* | *0.93* | *20.90* | *0.18* | *0.19* | *38.89* | *0.38* | *0.40* | *74.40* | *0.65* | *0.65* |
|  | *StdDev* | *1.63* | *0.02* | *0.01* | *4.49* | *0.08* | *0.08* | *4.79* | *0.05* | *0.04* | *10.17* | *0.07* | *0.07* |
| EQFREQ | Fold-1 | 53.33 | 0.62 | 0.54 | 50.75 | 0.40 | 0.34 | 52.78 | 0.54 | 0.53 | 91.20 | 0.74 | 0.74 |
|  | Fold-2 | 93.33 | 0.93 | 0.94 | 35.82 | 0.28 | 0.19 | 52.78 | 0.59 | 0.46 | 97.58 | 0.81 | 0.81 |
|  | Fold-3 | 100.00 | 1.00 | 1.00 | 29.85 | 0.35 | 0.19 | 41.67 | 0.27 | 0.42 | 99.94 | 0.90 | 0.90 |
|  | Fold-4 | 83.33 | 0.84 | 0.85 | 25.37 | 0.31 | 0.13 | 50.00 | 0.66 | 0.53 | 99.98 | 0.93 | 0.93 |
|  | Fold-5 | 83.33 | 0.81 | 0.83 | 32.84 | 0.37 | 0.23 | 47.57 | 0.65 | 0.53 | 99.97 | 0.87 | 0.87 |
|  | *Global Avg* | *82.67* | *0.84* | *0.83* | *34.93* | *0.34* | *0.22* | *48.96* | *0.54* | *0.49* | *97.73* | *0.85* | *0.85* |
|  | *Max* | *100.00* | *1.00* | *1.00* | *50.75* | *0.40* | *0.34* | *52.78* | *0.66* | *0.53* | *99.98* | *0.93* | *0.93* |
|  | *Min* | *53.33* | *0.62* | *0.54* | *25.37* | *0.28* | *0.13* | *41.67* | *0.27* | *0.42* | *91.20* | *0.74* | *0.74* |
|  | *StdDev* | *15.97* | *0.13* | *0.16* | *8.63* | *0.04* | *0.07* | *4.13* | *0.14* | *0.05* | *3.39* | *0.07* | *0.07* |
| GARST | Fold-1 | 100.00 | 1.00 | 1.00 | 52.24 | 0.46 | 0.35 | 83.33 | 0.84 | 0.84 | 96.80 | 0.97 | 0.97 |
|  | Fold-2 | 96.67 | 0.96 | 0.97 | 49.25 | 0.23 | 0.25 | 88.89 | 0.91 | 0.88 | 99.86 | 0.94 | 0.94 |
|  | Fold-3 | 90.00 | 0.90 | 0.90 | 43.28 | 0.40 | 0.24 | 69.44 | 0.69 | 0.66 | 99.96 | 0.93 | 0.93 |
|  | Fold-4 | 93.33 | 0.94 | 0.94 | 55.22 | 0.40 | 0.32 | 66.67 | 0.77 | 0.69 | 99.99 | 0.97 | 0.98 |
|  | Fold-5 | 96.67 | 0.97 | 0.97 | 56.72 | 0.42 | 0.38 | 68.06 | 0.76 | 0.72 | 99.99 | 0.94 | 0.94 |
|  | *Global Avg* | ***95.33*** | ***0.95*** | ***0.96*** | ***51.34*** | ***0.38*** | ***0.31*** | ***75.28*** | ***0.79*** | ***0.76*** | ***99.32*** | ***0.95*** | ***0.95*** |
|  | *Max* | *100.00* | *1.00* | *1.00* | *56.72* | *0.46* | *0.38* | *88.89* | *0.91* | *0.88* | *99.99* | *0.97* | *0.98* |
|  | *Min* | *90.00* | *0.90* | *0.90* | *43.28* | *0.23* | *0.24* | *66.67* | *0.69* | *0.66* | *96.80* | *0.93* | *0.93* |
|  | *StdDev* | *3.40* | *0.03* | *0.03* | *4.78* | *0.08* | *0.05* | *9.06* | *0.07* | *0.09* | *1.26* | *0.02* | *0.02* |

Figure 5 describes the distribution of the accuracy values for each test. From this figure, it can be seen that the GARST method produces consistent accuracy values, although it is not always superior. Thus, it can be concluded that the GARST method is generally proven to have a superior performance in terms of accuracy and reliability, as measured by precision and recall, compared to the other two methods.

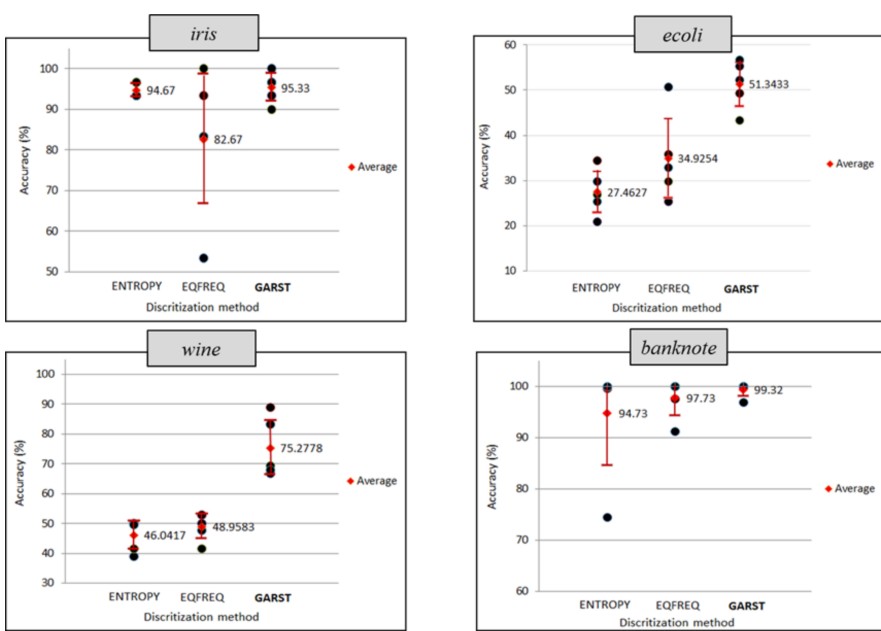

**Figure 5.** Plots showing the distribution of accuracy values.

According to non-parametric statistical testing, namely the Friedman test, as shown in Table 5, the *p*-value obtained is smaller than 0.05, so it can be concluded that there is a significant difference between the three methods. Meanwhile, from the post hoc test results, as shown in Table 6, the *p*-values of ENTROPY vs. GARST and EQFREQ vs. GARST are all less than 0.05, so it can be concluded that the GARST method is a method that has a significant difference compared to the other two methods.

**Table 5.** The results of Friedman test for the accuracy.

| Dataset | Discretization Methods | | |
| :---: | :---: | :---: | :---: |
| | **ENTROPY** | **EQFREQ** | **GARST** |
| *iris* | 96.67 | 53.33 | 100.00 |
| | 93.33 | 93.33 | 96.67 |
| | 96.67 | 100.00 | 90.00 |
| | 93.33 | 83.33 | 93.33 |
| | 93.33 | 83.33 | 96.67 |
| *ecoli* | 29.85 | 50.75 | 52.24 |
| | 34.33 | 35.82 | 49.25 |
| | 26.87 | 29.85 | 43.28 |
| | 20.90 | 25.37 | 55.22 |
| | 25.37 | 32.84 | 56.72 |
| *wine* | 38.89 | 52.78 | 83.33 |
| | 50.00 | 52.78 | 88.89 |
| | 41.67 | 41.67 | 69.44 |
| | 50.00 | 50.00 | 66.67 |
| | 49.65 | 47.57 | 68.06 |
| *banknote* | 74.40 | 91.20 | 96.80 |
| | 99.54 | 97.58 | 99.86 |
| | 99.89 | 99.94 | 99.96 |
| | 99.88 | 99.98 | 99.99 |
| | 99.96 | 99.97 | 99.99 |
| Friedman Test Result | *p-value* | 0.000003224 | |

**Table 6.** The results of post hoc test.

| Method | ENTROPY | EQFREQ | GARST |
| :---: | :---: | :---: | :---: |
| **ENTROPY** | 1.000 | 0.556 | **0.001** |
| **EQFREQ** | 0.556 | 1.000 | **0.001** |
| **GARST** | **0.001** | **0.001** | 1.000 |

## 7. Conclusions

A method to improve the accuracy and reliability of the RST-based classifier model has been proposed by involving the RST instruments at the beginning of the discretization process. This method uses a *k*-means-based discretization method optimized with a genetic algorithm (GA). As a result, the method was proven not to sacrifice the degree of information quality from the dataset and the performance was quite competitive compared to the popular *state-of-the-art* methods, namely *equal-frequency* and *entropy-based*. Moreover, the proposed discretization method based on k-means optimized by GA and using one of the rough set theory instruments has proven to be effective for use in the RST classifier.

The test of the discretization method proposed in this study uses four datasets that have different profiles in the *5*-fold scenario, and the results were tested by using Friedman and post hoc tests; therefore, it can be concluded that the proposed method should be effective for discretization purposes to any dataset, especially for the RST-based classification cases. The disadvantage of this proposed method is an unstable speed during

discrete processes, especially in the optimization of the number of bins. This is due to the application of a heuristic approach by GA.

**Author Contributions:** Writing—review & editing, T.H.D., N.A.S. and T.B.A. All authors have read and agreed to the published version of the manuscript.

**Funding:** This research received no external funding.

**Institutional Review Board Statement:** Not applicable.

**Informed Consent Statement:** Not applicable.

**Data Availability Statement:** All the datasets used in this study were taken from the UCI public data repository.

**Conflicts of Interest:** The authors declare no conflict of interest.

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
