# Peer review of "Rough-Set-Theory-Based Classification with Optimized k-Means Discretization"

_technologies, doi:10.3390/technologies10020051_

Round 1

Reviewer 1 Report

Line 33: Sentence is wrong and largely 1st paragraph can you explain better.
Line 52: Any references or citations for not optimal here?
Line 54: Sentence is wrong
Line 64: Paragraph needs revision
The sample size, features are very small. What is the % of continuous attributes in the dataset?
I feel lot more citations missing on existing work/literature.
I feel overall writing quality and language errors should be improved.

Author Response

Dear Reviewer 1,

Thank you very much for the valuable feedbacks you gave.  In general, I accept and agree with your feedback, and I have done the advice you gave. I am currently submitting the manuscript for proofread. My point-by-point responses to your feedbacks can be seen in the uploaded attachment. 

Thank you very much.

Regards,

Teguh H. Dwiputranto

Reviewer 2 Report

Main objective of this research is to propose a method to improve the accuracy and reliability of the Rough Set Theory based classifier model by involving RST instruments at the beginning of the discretization process.

This paper may be accepted subject to a minor revision.

  1. The motivation of this study is not clearly illustrated. You should make a clear discussion about the literature and show the challenges that need to be tackled.
  2. Are there any limitations to the proposed model? Please mention this in the conclusion section.
  3. The future research scope is not clearly mentioned in the conclusion section. More specific future research directions are welcome. 
  4. I strongly suggest authors to use simple symbols. The abstract needs to be condensed.
  5. Some comments on literature are not up to date. Please update literature related to q-Rung orthopair m-polar rough set, Pythagorean m-polar rough set, m-polar n and Linear Diophantine fuzzy rough sets, etc.
  6. A comparative analysis of the suggested technique with existing approaches is missing. Add advantage and limitations of proposed work.
  7. Please missing citations especially in Definitions from existing literature.
  8. No illustration is section 2 is given. Add illustrations of proposed notions.
  9. Figure 3. Pseudo-code of the proposed method is not clear.

Author Response

Dear Reviewer 2,

Thank you very much for the valuable feedbacks you gave.  In general, I accept and agree with your feedback, and I have done the advice you gave. I am currently submitting the manuscript for proofread. My point-by-point responses to your feedbacks can be seen in the uploaded attachment. 

Thank you very much.

Regards,

Teguh H. Dwiputranto

Reviewer 3 Report

This submission approaches a classification problem. The technique, that relies on rough set theory, begins with the discretization of continuous attributes through a k-means-based methodology optimized with genetic algorithm. Tests are applied that confirm that the new technique outperforms two other discretization methods.

Comments for improvement.

1. Simplify the Abstract. It gives too much technical information about the contribution. Abstracts should be concise and to the point, avoiding unnecessary details. Its organization can be improved too.

2. Correct and proofread the manuscript. Concerning the Abstract only: it should keep the present tense, and sentences like “to measure the significant of the performance different.” contain various errors. Concerning the  first paragraph of Introduction: “researchers commonly done” is wrong, “the method such as” should probably be “methods such as”.

3. Some examples would help understand so many concepts in section 2. Technically, the paper lacks formal accuracy. Below are some reasons from section 2:

Decision tables, or information tables: what are they?

Consider sections 2.3 and 2.4: what is C? and, what is the information that defines the two approximations: is it an equivalence relation, or an information system? (my guess: it is an information system but we do not know its components).

So without this information, it is difficult to assess if there are typos in eq. (1) and (2): I would say that the equivalence class of x, not X, is needed, that is to say, [x]_B should replace [X]_B. Check line 126 for a similar issue.

Probably there is a typo in eq. (3) too: BN_B(X).

Although section 2.7 defines “crisp set”, I believe that a standard terminology is definable set.

Section 2.8 uses U/D which has not been defined before.

Eq. (6) is not accurate.

I do not understand Eq. (8).

4. The presentation of Algorithm 1 is deceptive. Can't you produce a more readable output? Now it appears to be a blurry screen capture.

Author Response

Dear Reviewer 3,

Thank you very much for the valuable feedbacks you gave.  In general, I accept and agree with your feedback, and I have done the advice you gave. I am currently submitting the manuscript for proofread. My point-by-point responses to your feedbacks can be seen in the uploaded attachment. 

Thank you very much.

Regards,

Teguh H. Dwiputranto

Round 2

Reviewer 1 Report

Overall the article looks good after the authors revised it.

Reviewer 3 Report

I have read the responses and I am satisfied with the current version. Thank you. I have no further comments on R1.